# RUNX1-Regulated Signaling Pathways in Ovarian Cancer

**DOI:** 10.3390/biomedicines11092357

**Published:** 2023-08-23

**Authors:** Yuanzhi Chen, Yingying He, Shubai Liu

**Affiliations:** 1State Key Laboratory of Phytochemistry and Plant Resources in West China, Kunming Institute of Botany, Chinese Academy of Sciences, Kunming 650201, China; chenyuanzhi@mail.kib.ac.cn; 2University of Chinese Academy of Sciences, Beijing 100049, China; 3School of Chemical Science & Technology, Yunnan University, Kunming 650091, China

**Keywords:** Runt-related transcription factor 1 (RUNX1), ovarian cancer, apoptosis, signaling pathways, clinical therapeutic target

## Abstract

Ovarian cancer is the leading cause of gynecological death worldwide, and its poor prognosis and high mortality seriously affect the life of ovarian cancer patients. Runt-related transcription factor 1 (RUNX1) has been widely studied in hematological diseases and plays an important role in the occurrence and development of hematological diseases. In recent years, studies have reported the roles of RUNX1 in solid tumors, including the significantly increased expression of RUNX1 in ovarian cancer. In ovarian cancer, the dysregulation of the RUNX1 signaling pathway has been implicated in tumor progression, metastasis, and response to therapy. At the same time, the decreased expression of RUNX1 in ovarian cancer can significantly improve the sensitivity of clinical chemotherapy and provide theoretical support for the subsequent diagnosis and treatment target of ovarian cancer, providing prognosis and treatment options to patients with ovarian cancer. However, the role of RUNX1 in ovarian cancer remains unclear. Therefore, this article reviews the relationship between RUNX1 and the occurrence and development of ovarian cancer, as well as the closely regulated signaling pathways, to provide some inspiration and theoretical support for future research on RUNX1 in ovarian cancer and other diseases.

## 1. Introduction

Ovarian cancer is one of the most intractable malignant tumors of the female reproductive system, and its difficulties in diagnosis and high fatality rate have not been overcome. Surgery, platinum-based chemotherapy, and maintenance therapy are the main therapeutic methods. At the time of diagnosis, most patients are in the middle and late stage, with extensive metastasis, and the 5-year survival rate is only 47.6%. According to the World Health Organization, an estimated 225,500 cases of ovarian cancer are diagnosed each year, and 140,200 patients will die from the disease, making it the seventh most common form of cancer worldwide and the eighth leading cause of cancer-related death [1,2]. From the overall perspective of early diagnosis and treatment of ovarian cancer, experts and scholars around the world have made significant progress, including targeted therapy and immunotherapy, but further research is still needed to identify the etiology to screen novel and effective therapeutic targets and formulate plans for early detection and prevention.

Runt-related transcription factor 1 (RUNX1) is a transcription factor in the RUNX family of transcription factors that play a vital role in a wide range of biological processes including cell proliferation, apoptosis, differentiation, and lineage determination [3,4]. By directly activating or inhibiting the expression of target genes, RUNX1 plays a crucial role in early development and cell-line specification in various tissues [5]. Thus, the misregulation of RUNX1 is associated with cancer. For example, RUNX1 is necessary for normal hematopoietic production [6], and genetic alterations in the *RUNX1* gene have been linked to various forms of leukemia and other hematological malignancies [7]. Although misregulation of RUNX1 was first identified in blood-related cancers, many researchers have now identified its role in many different types of cancer, especially tumors of epithelial origin [8]. Meanwhile, RUNX1 disorders have been strongly linked to breast, uterine, ovarian, and other female-related cancers [9,10]. RUNX1 expression features in normal issues and cancers, and the mechanism of action of RUNX1 in many cancer types has highlighted the potential of RUNX1 as a biomarker for cancer. However, the mechanism of RUNX1, especially in ovarian cancer, is still largely unknown. Therefore, this review summarizes the reported signaling pathways involved in regulating RUNX1, to provide some help for follow-up research on and even treatment of ovarian cancer.

## 2. Expression and Regulation of RUNX1 in Ovarian Cancer

Although ovarian cancer does not feature the highest morbidity and mortality, its morbidity and mortality are very high among gynecological tumors. The formation of ovarian tumors may be closely related to environmental and/or genetic factors. As a transcription factor of the RUNX transcription factor family, RUNX1 plays a vital role in the occurrence and development of various diseases through its ability to directly activate or inhibit the expression of target genes [5]. Therefore, changes in RUNX1 expression are also closely related to cancer. Although studies on RUNX1 were initially explored in blood-related cancers, its roles has now been gradually reported and identified in studies of many solid tumors, especially those of epithelial origin. RUNX1 can function as an oncogene or a tumor suppressor gene, depending on the type of cancer. Among different types of cancer, changes in RUNX1 expression are associated with female-related cancers such as breast cancer, uterine cancer, and ovarian cancer [9,10] (Figure 1). Genetic alterations in *RUNX1* have previously been identified in 1.5% of ovarian cancers, according to publicly available data from The Cancer Genome Atlas (TCGA). RUNX1 is associated with various types of cancer, especially hematological malignancies and cancers of epithelial origin. In mice, RUNX1 is involved in early ovarian development and is expressed in granulosa cells and the ovarian surface epithelium of embryonic ovaries [11]. RUNX1 is also overexpressed in human advanced (metastatic) EOC, possibly due to an epigenetic mechanism associated with DNA hypomethylation in its promoter region [12]. At the same time, it has been reported that RUNX1 is upregulated in human ovarian epithelial cancer (EOC) tissues and is related to the proliferation, migration, and invasion of EOC cells [10]. RUNX1 promotes the phosphorylation of the transcription factor STAT3 with oncogenic properties through the JAK/STAT pathway, thereby inhibiting p21 and promoting STAT3 activation [13,14]. Phosphorylation of STAT3 at its conserved tyrosine residues promotes its homodimerization and translocation to the nucleus, thereby regulating the transcription of genes involved in survival, proliferation, and invasion, and constitutive activation of STAT3 is highly associated with malignant transformation of cancer and poor clinical outcome [15,16]. In summary, RUNX1 plays an essential role in the development of ovarian cancer and can regulate the function of ovarian cancer through multiple pathways. Therefore, it is crucial to further elucidate the mechanism of RUNX1 in ovarian cancer.

## 3. Interaction between RUNX1 and Its Cofactors

RUNX1 forms complexes with various cofactors to regulate gene expression. In ovarian cancer, the interaction of RUNX1 with cofactors such as CBFB (core-binding factor beta) and SMAD proteins has been reported [17]. These interactions modulate the transcriptional activity of RUNX1 and affect downstream signaling. Genetic alterations in CBFB and RUNX1 are associated with many types of human diseases and cancers. At the molecular level, RUNX1 and CBFB form a transcriptional complex. The putative mechanism of action of the CBFB/RUNX1 complex is that RUNX1 is a sequence-specific DNA-binding transcription factor. In contrast, CBFB has no DNA-binding activity, but it heterodimerizes with RUNX1 in the nucleus and enhances the DNA-binding and transcriptional activities of RUNX1 [18,19]. RUNX1 and CBFB form the CBFB/RUNX1 transcription complex, which regulates the transcription of many genes, such as NOTCH3 [20]. Previous studies have shown that RUNX1 transcription factors associate with co-activators CBP and p300 [21], whereas leukemia-associated RUNX1-ETO proteins do not interact with CBP/p300 family proteins, but rather with co-repressive proteins including HDACs. Therefore, it has been proposed that RUNX1-ETO drives aberrant gene activity by recruiting repressive chromatin modifiers to activate promoters [22]. One study found that RUNX1-ETO is essential for maintaining cyclin D2 (CCND2) expression [23]. In summary, RUNX1 interacts with its cofactors to activate the transcriptional activity of RUNX1 and further regulate the downstream target genes and related signaling pathways. This further demonstrates the importance of RUNX1 in ovarian cancer and justifies the need for in-depth research on RUNX1.

## 4. Regulation of Cell Proliferation and Survival by RUNX1

Human cancers have properties known as signatures, of which continued proliferation, apoptosis evasion, and genomic instability are the most prevalent. The normal proliferation function of tumor cells is crucial, and the continuous proliferation ability is also one of the significant characteristics of tumor cells. It is significant to find the key factors regulating cell proliferation for cancer treatment. Studies have reported that RUNX1 can regulate the proliferation of various solid tumors. RUNX1 is a target gene of miR-18a-5p and inhibits the proliferation, migration, and invasion of malignant melanoma cells [24]. High-throughput sequencing of glioblastoma cohorts identified BCL3, COL3A1, MGP, and MXI1 as potential RUNX1 target genes that promoted cell proliferation, invasion, and adhesion in a TGFβ pathway-dependent manner [25]. RUNX1 regulated the RUNX1/p-SMAD3/SUV39H1GBM signaling pathway to affect proliferation and invasion in a TGFβ pathway-dependent manner [25]. Overexpression of CENPE, the direct target gene of the so-called RUNX1-deficient phenotype is a possible mechanism underlying the RUNX1–CENPE axis, which promotes leukemic cell growth [26]. As a transcription factor, RUNX1 may affect tumor proliferation by regulating target genes. Meanwhile, RUNX1 can regulate the proliferation of ovarian cancer through the TGFβ signaling pathway. These results suggest that RUNX1 may be involved in the complex regulatory network in ovarian cancer, and its regulation of signaling pathways in ovarian cancer still needs further study.

Apoptosis is essential for development and is associated with a variety of diseases, including autoimmune diseases, neurodegenerative diseases, bacterial and viral diseases, heart diseases, and cancer progression [27,28]. In leukemia cells, studies have found a corresponding regulatory relationship between RUNX1 and BCL2, in which BCL2 as an essential regulator participates in the dual function of RUNX1 on survival [29]. Although RUNX1 recognizes the consensus binding sequence (TGTGGT) in the BCL2 promoter, it requires ETO in conjunction with RUNX1 to generate the RUNX1/ETO fusion protein that activates transcription of the BCL2 gene through its promoter [30]. Therefore, RUNX1 cannot directly regulate the activity of the BCL2 promoter, and the investigators found that RUNX1 can regulate BCL2 through the miR-17–92 cluster in ovarian cancer, which may be part of the mechanism via which RUNX1 regulates BCL2 to further regulate apoptosis [31]. Moreover, the transcription factor RUNX1 was discovered to restrain cisplatin-induced apoptosis in ovarian cancer cells [31]. RUNX1 plays an essential role in the regulation of apoptosis, suggesting the therapeutic potential of targeting RUNX1 in various diseases, especially cancer, and the development of novel RUNX1-based therapeutic approaches may be possible.

## 5. Regulation of P53 by RUNX1

As a nuclear transcription factor, tumor suppressor p53 plays a vital role in the occurrence and development of tumors, and it also participates in the regulation of many pathways in cancer [32]. P53 is activated in response to stress and can regulate cancer cells by regulating signal transduction. P53 can be activated by a variety of stress signals, including genotoxic damage, oncogene activation, nutrient starvation, and hypoxia [33,34,35]. Recently, a study showed that p53 can accumulate in the cytoplasm of HGSOC, further affecting its function [7]. Furthermore, an increasing number of studies have also shown that RUNX proteins act as regulators of the DNA damage response, often in conjunction with the p53 pathway [36,37,38]. Another study reported that p53 directly transactivates the CBFB promoter through its p53 response element-like sequence, and RUNX1 may indirectly regulate CBFB expression through p53 [38]. Meanwhile, the RUNX1–p53–CBFB regulatory circuit contributes to the acquisition of drug resistance. Some studies have shown that RUNX1 expression is increased in pathological cardiac hypertrophy [39]. RUNX1 gain- and loss-of-function studies performed in vitro showed that RUNX1 promoted cardiomyocyte hypertrophy. In addition, knockdown of RUNX1 expression alleviated pathological cardiac hypertrophy in vivo. Mechanistically, RUNX1 binds to the p53 gene and promotes its transcription [40]. RUNX1 acts as a transcription factor that binds to the p53 gene and promotes its expression. In summary, RUNX1 can act as a key factor in the regulation of p53 and further regulate other physiological functions. Therefore, it is necessary to further study RUNX1.

## 6. Regulation of Epithelial–Mesenchymal Transition (EMT) by RUNX1

EMT plays an important role in the occurrence and development of many diseases, and its basic process is reactivated during cancer progression. Transforming growth factor β (TGFβ) regulates biological processes such as cell growth, differentiation, apoptosis, and homeostasis [41,42]. Meanwhile, TGFβ has been reported to be the most important inducer of epithelial-to-mesenchymal transition (EMT) [43,44]. Studies have shown that TGFβ can regulate EMT by inducing RUNX1 expression in renal tubular epithelial cells. Previous studies have demonstrated that RUNX1 is an important downstream factor of the TGFβ signaling pathway and can regulate TGFβ-induced migration and EMT in CRC cells [45,46]. TGFβ can induce RUNX1 expression, and RUNX1 knockout can inhibit TGFβ-induced migration and EMT of colorectal cancer cells [46]. RUNX1 expression in CRC is mediated by Smad activation following TGFβ stimulation. RUNX1 is reported to be a key regulator in the TGFβ signaling pathway that prevents the progression of HCC [47], and previous studies have suggested that RUNX1 may play an important role in regulating EMT by interacting with the intracellular TGFβ/Smad signaling pathway [48,49,50]. RUNX1 is highly expressed in ovarian cancer, and its expression level is dependent on Smad3. RUNX1 can also promote the activation of TGFβ/Smad2/3/4 signaling [51]. The above indicates that RUNX1 also plays an important role in regulating TGFβ signaling pathway, mediating downstream signaling pathways to regulate cell physiological functions by regulating TGFβ. It also further demonstrates the potential for further research on RUNX1.

Epidermal growth factor receptor (EGFR) is a member of the ErbB receptor tyrosine kinase family, which plays a crucial role in cell growth, differentiation and movement. Meanwhile, EGFR also plays an important role in the development of EMT. EGFR regulates EMT-related proteins via AKT and influences tumor cell function. However, it has been reported that EGFR can regulate the downstream factor STAT3, and RUNX1 knockdown can significantly affect the expression of EGFR; thus, RUNX1 may affect the expression of STAT3 by affecting EGFR [52]. The transcription factor RUNX1 acts as a downstream effector of mTORC1 and upregulates EGFR at the transcriptional level by directly binding to the promoter of the EGFR gene. One signaling pathway, RUNX1/EGFR/STAT3, was found to drive EMT development and carcinogenesis through dysregulated mTORC1 [53]. Therefore, RUNX1 and STAT3 inhibitors are also expected to have therapeutic value in the treatment of mTORC1-related cancers [52]. Taken together, RUNX1, as a very important STAT3 regulator, may act as a signal transduction factor through EGFR, which provides a theoretical basis for further understanding the roles of RUNX1 in ovarian cancer.

Signal transducer and activator of transcription 3 (STAT3) is a cytoplasmic transcription factor that regulates cell proliferation, differentiation, apoptosis, angiogenesis, inflammation, and immune response [54,55,56]. In many human tumors, aberrant activation of STAT3 triggers tumor progression through oncogene expression, leading to increased malignancy of the tumor; therefore, targeting STAT3 may improve tumor progression and, thus, suppress tumor immune responses. RUNX1 appears to be essential for solid tumor initiation and maintenance through upstream stimulation of STAT3 signaling, a central cancer pathway associated with a variety of cancer diseases [57,58,59,60]. Activated STAT3 is present in the vast majority of human oral cancers, as well as breast, lung, ovarian, pancreatic, and prostate cancer, myeloma, leukemia, and lymphoma [58]. The activity of STAT3 is regulated by RUNX1, and the loss of RUNX1 leads to the inhibition of STAT3 phosphorylation activity [61,62]. In addition, overexpression of RUNX1 leads to STAT3 activation and is required for the regulation of skin cell proliferation and tumorigenesis [13]. TGF3 is a key target of RUNX1 signaling, and this signaling axis can be mediated by STAT3 activation. Interestingly, the expression of SOCS3, an inhibitor of STAT3, was specific and upregulated by RUNX1 deficiency. This study reported a novel regulatory pathway, the RUNX1–STAT3–TGF3 axis [63], which provides support for further understanding of the signaling pathways regulated by RUNX1 in ovarian cancer. Furthermore, the defective phenotype of RUNX1 mutants was rescued by treatment with STAT3 activators, suggesting that STAT3 phosphorylation regulates epithelial stem cells, thereby mediating the RUNX1–LGR5 axis to regulate the homeostasis of continued incisor growth [64]. Taken together, RUNX1 is involved in the regulation of STAT3 and its downstream signaling pathways in many diseases. Meanwhile, RUNX1 can act as a key factor in the regulation of EMT and further promote the invasion and metastasis of cancer cells by EMT. It further highlights the potential of RUNX1 in the treatment of ovarian cancer and proves the necessity of in-depth research on RUNX1.

## 7. Potential of RUNX1 in Clinical Therapy

For HGSOC, first-line treatment is surgery with disease staging followed by cytotoxic chemotherapy with platinum and taxanes; ovarian cancer that recurs within 6 months of completion of initial treatment is considered platinum-resistant. Targeted therapy uses agents that target pathways that are critical for the growth or survival of ovarian cancer cells. These drugs target specific proteins, reducing the adverse effects on normal cells that often occur with cytotoxic chemotherapy, and they can be largely used to treat patients who are resistant to chemotherapy drugs [65]. Therefore, it is necessary to further investigate the pathways that are critical for the regulation of ovarian cancer cell growth or survival.

Targeted therapy for ovarian cancer most classically involves molecules that inhibit angiogenesis. As tumors grow, they tend to become hypoxic, which triggers the formation of new blood vessels. The signaling between vascular endothelial growth factor (VEGF) and its receptor (VEGFR) can promote the proliferation and migration of endothelial cells [66]. Previous studies revealed that linc00174 epigenetics inhibited ZNF24 and RUNX1 expression by binding EZH2, thereby reducing the inhibition of VEGFA and promoting angiogenesis [67]. Abu-Jawdeh et al. showed that angiogenesis plays a role in the development of ovarian cancer [68]. Bevacizumab, which is widely used in clinical targeted therapy, is a monoclonal antibody against VEGFA. At the same time, bevacizumab is also widely used in the initial and recurrent treatment of platinum-sensitive and platinum-resistant ovarian cancer. In clinical practice, bevacizumab can also be used in combination with conventional chemotherapy drugs (cisplatin, paclitaxel, etc.), presenting a better therapeutic effect on patients with conventional chemotherapy drug resistance. This further suggests that inhibition of important factors in the development of ovarian cancer combined with conventional clinical drugs may have a better therapeutic effect.

Although paclitaxel and platinum agents (carboplatin or cisplatin) are widely used alone or in combination to treat ovarian cancer, the development of chemotherapy resistance significantly reduces the efficacy of these agents [69]. However, most patients treated with taxanes do not exhibit complete destruction of tumor cells; resistance and relapse to taxanes are frequently observed, ultimately leading to recurrence and poor prognosis, significantly limiting the impact of taxane-based strategies on ovarian cancer treatment [70,71]. At the same time, RUNX1 is overexpressed in ovarian cancer cells, which activates the NF-κB pathway and leads to chemotherapy resistance. NF-κB inhibitor significantly inhibits the proliferation of drug-resistant ovarian cancer cells and induces their apoptosis. Interestingly, one study found that metformin combination therapy significantly reduced cell proliferation and migration by increasing chemotherapy sensitivity to cisplatin and paclitaxel, and by reducing ERK and AKT kinase activation [72]. According to previous reports, RUNX1 deletion combined with chemotherapy drugs (taxanes, paclitaxel, and platinum) would have a better effect on chemotherapy resistance in ovarian cancer. In our study, knockdown of RUNX1 significantly increased the proliferation inhibition of ovarian cancer cells treated with taxane, paclitaxel, and platinum drugs (carboplatin or cisplatin). Therefore, for ovarian cancer patients with poor prognosis and drug resistance, reducing or inhibiting RUNX1 gene expression in ovarian cancer combined with cisplatin therapy may have a significant therapeutic effect as a combination treatment strategy.

## 8. Prospects

Ovarian cancer is a serious cancer disease; although its incidence is not high in the global population, it still has the highest mortality of gynecological malignant tumors. Data from the World Health Organization highlight the importance of ovarian cancer in contributing to morbidity and mortality in populations worldwide. Therefore, it is particularly important and urgent to find an effective treatment for ovarian cancer. At present, the clinical treatment of ovarian cancer mainly involves chemotherapy and radiotherapy, and even surgical resection of the ovary. In particular, patients who have already undergone surgery at the time of diagnosis, as well as those who cannot undergo surgery because of metastasis, are required to receive chemotherapy. This is a major threat to women’s health worldwide. Since long-term chemotherapy can cause tumor cells to become resistant to a single chemotherapeutic agent, resulting in poor results after chemotherapy, the idea of drug combination has been proposed. Chemotherapy drugs play an important role in the treatment of ovarian cancer; however, due to their cytotoxicity, they also seriously damage the normal cells of the patient, leading to many side-effects [73,74]. As a result, patients must endure psychological stress and great physical discomfort during treatment, which further leads to poor quality of life.

Runt-related transcription factor 1 (RUNX1), also known as acute myeloid leukemia 1, is a member of the RUNX transcription factor family. Its evolutionary conservation makes it play a key determinant role in various disease tissues [75]. *RUNX1* is one of the most frequently mutated gene in a variety of hematological malignancies and plays a tumor suppressor role in leukemia. At the same time, RUNX1 also acts as an oncogene and tumor suppressor gene in epithelial tumors, and its role in various cancers has attracted more and more attention [76,77]. RUNX1 is overexpressed in various epithelial tumors, especially in the tumor initiation stages. Moreover, genetic alterations in RUNX1 also occur in many solid tumors [78]. Several studies on ovarian cancer have found that RUNX1 can promote cell proliferation, migration, and invasion of ovarian cancer [10,79]. Similar results were found in our study. In addition, RUNX1 was shown to promote cell proliferation and tumor growth in esophageal cancer [80]. This indicates that RUNX1 is closely related to the occurrence and development of ovarian cancer, as well as proves the necessity of further in-depth study of RUNX1 in ovarian cancer.

Previous studies have reported that RUNX1 is significantly highly expressed in ovarian cancer and can also regulate many functions of tumor cells, indicating that RUNX1 is crucial in the regulation of ovarian cancer [10,79]. At the same time, RUNX1 also participates in the regulation of multiple signaling pathways (Figure 2). One study reported that RUNX1 knockdown or deletion in the absence of p53 could significantly inhibit tumor initiation and progression with better prognosis [81]. At the same time, RUNX1 can bind to the p53 gene and promote its expression, thereby mediating the pathological process by regulating p53 [40]. RUNX1 has also been reported to be involved in TGFβ-induced epithelial–mesenchymal transition (EMT) in a Smad3-dependent manner [45]. Meanwhile, Lu et al. demonstrated that RUNX1, as an important downstream factor of TGF-β signaling pathway, can regulate colon cancer cell migration and EMT in colon cancer [46]. Furthermore, the loss of RUNX1 affects the expression of EGFR and STAT3, thereby regulating the downstream pathways and ultimately regulating the function of tumor cells [82]. In addition to the above reports that are relatively clear, there are also other signaling pathways involved in the regulation of RUNX1, such as the Wnt signaling pathway [53,83], Notch1 signaling pathway [84], MAPK signaling pathway [85], and NF-κB signaling pathway [86]. These reports further suggest that RUNX1 may be involved in the transduction of multiple signaling pathways, thereby participating in the regulation of the development of ovarian cancer. In summary, RUNX1 can regulate multiple pathways to regulate the function of tumor cells. At the same time, it also provides a certain direction and theoretical basis for further research on the function of RUNX1.

Although considerable progress has been made in clinical chemotherapy, the 5-year survival rate of ovarian cancer patients is still less than 50%, which is mainly due to chemotherapy resistance [87]. Primary and acquired platinum resistance confer a serious poor prognosis for patients with ovarian cancer. Therefore, it is essential to find a clinical solution as soon as possible. The mechanisms of chemoresistance of ovarian cancer cells to chemotherapy drugs are complex, including DDR, oxidative stress, cell-cycle regulation, cancer stem cells, immunity, apoptosis pathway, autophagy, and abnormal signaling pathways. Therefore, a single clinical mechanism has not been able to have a better effect on patients with ovarian cancer, and combination therapy with therapies targeting key factors in ovarian cancer may have a better clinical effect. In our previous study, the clinical chemotherapeutic drug 5-fluorouracil had a stronger inhibitory effect on the proliferation of colon cancer cells in the absence of potential colon cancer biomarkers BGN and THBS2 [88]. It was also found in our study that depletion of RUNX1 could significantly enhance the inhibitory effect of chemotherapy drugs on ovarian cancer cells. These results indicate that the key factors in the process of tumor development have the potential to be combined with clinical chemotherapy drugs for targeted therapy. They also further indicate that RUNX1 has the potential to be combined with chemotherapy drugs in the clinical treatment of ovarian cancer, thereby bringing better prognosis and quality of life to patients with ovarian cancer (Figure 3).

In conclusion, the transcription factor RUNX1 plays an important role in multiple solid tumors and is highly expressed in ovarian cancer, in which it plays a key role, indicating that RUNX1 has the potential as a biomarker for the diagnosis and treatment of ovarian cancer. At the same time, RUNX1 is also involved in the regulation of multiple signaling pathways, highlighting its key role in the development of tumors. These findings indicate that RUNX1 has the potential to be further studied in ovarian cancer, as well as the potential to be combined with clinical chemotherapy drugs. It is worth noting that the RUNX1 pathway in ovarian cancer is still an active area of research, and further studies are required to fully elucidate its molecular mechanisms and therapeutic potential.

## Figures and Tables

**Figure 1 biomedicines-11-02357-f001:**
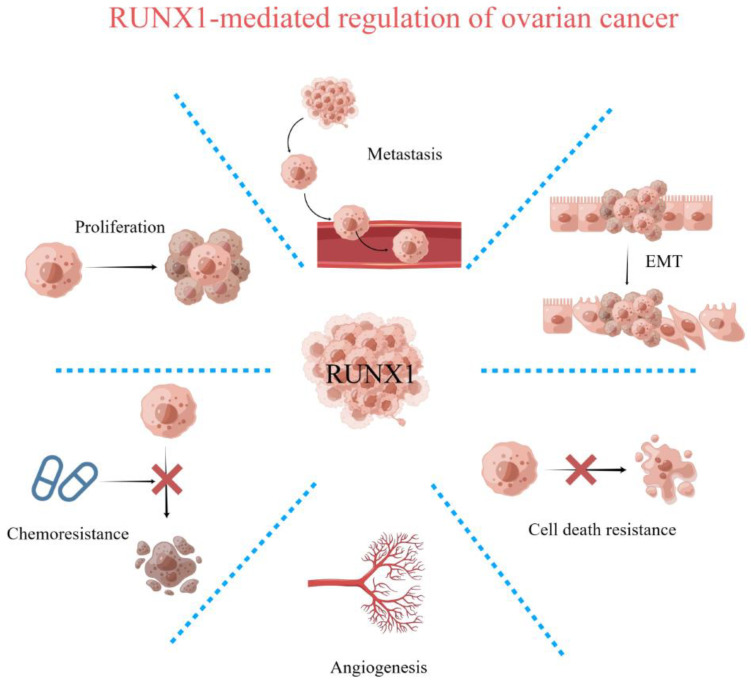
Regulation of key functions by RUNX1 in ovarian cancer.

**Figure 2 biomedicines-11-02357-f002:**
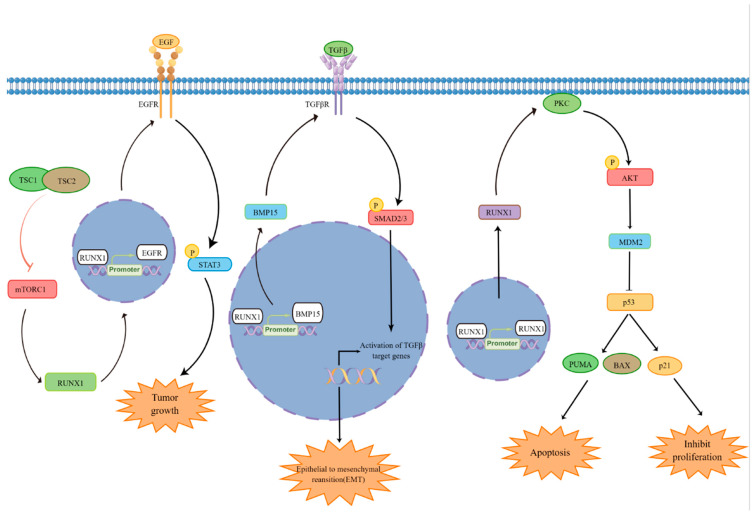
The molecular mechanisms of RUNX1 involved in apoptosis, EMT, and other functions. At present, it is clearly reported that RUNX1 is involved in regulating the EGFR signaling pathway, TGF-β signaling pathway, and p53 signaling pathway.

**Figure 3 biomedicines-11-02357-f003:**
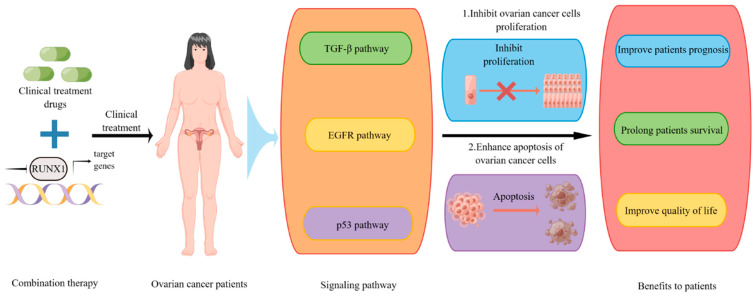
Prospects of inhibition of RUNX1 expression or transcription in combination with clinical treatment of ovarian cancer drugs is used in the clinical treatment of ovarian cancer.

## Data Availability

Not applicable.

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
