# Peer review of "RUNX1-Regulated Signaling Pathways in Ovarian Cancer"

_biomedicines, 2023, doi:10.3390/biomedicines11092357_

Round 1

Reviewer 1 Report

In this review article, the authors provided an overview of RUNX1 in ovarian cancer. Overall, the information is important, however, more work needs to be put in in order to understand the current advance of RUNX1 in ovarian cancer.

1   (1) Some similar descriptions were mentioned in more than one section, such as background information of RUNX1, STAT3, and ovarian cancers. You may want to keep the review as concise as possible by removing the redundant parts and reorganizing the paper.

2   (2) More papers need to be included to clearly display the link between RUNX1 and ovarian cancer.

3    (3) Section 2:  this section can be moved to the last section before prospect and can be the section which is all about RUNX1 in ovarian cancer.  

4    (4) Prospects: some information in this section haven been described in other sections.  This section can be more focused on prospects instead of background information. Try to keep this section more concise.  

     (5)  More detail can be described in the figure legends.

Author Response

1: The repeated contents in several sections have been deleted and further elaborated.

2: Some inconsistent or inappropriate references were deleted in the revision, and new references were added in the added content.

3: A reply to the reviewer's proposal to adjust the position of section 2: the author's section 2 is to explain that RUNX1 is significantly highly expressed in ovarian cancer to prove that RUNX1 influences key functions in ovarian cancer, so as to better transition to the following review of signaling pathways.

4: The same sections in prospects as described in other sections have been cut or streamlined accordingly.

5: Appropriate adjustments have been made to the expression of legends.

Reviewer 2 Report

The manuscript by Yuanzhi Chen et al. aimed to report the information regarding the expression and role of the transcription factor RUNX-1 in ovarian cancer. Not many publications can be found in Pubmed specifically reporting data about RUNX1 in ovarian cancer. Besides those cited in this manuscript there are other references which could have been mentioned. As an example:  

PLoS One. 2022 Jul 22;17(7):e0271539. doi: 10.1371/journal.pone.0271539. eCollection 2022.

The manuscript reports data that can be considered of interest but the data could have been better organized. While a lot of references have been cited, many mistakes in the citations have been found to make this review more appealing for researchers working on ovarian cancer.

The topic related to ovarian cancer, such as regulation of proliferation, interaction with p53 or EGFR/STAT3 signaling, are fine but the Prospects section somehow contains the main information but at the end it only appears as a lot of repetition.

Overall, it is suggested to better organize this manuscript, maybe starting with the general information about RUNX-1 target genes and biological function in leukemia, then in solid tumors thus going to ovarian cancer. Accordingly, the presented figures could be inserted within the text.

·           Following, ‘some’ discrepancies about citations are reported:

#1,2,19, #9 at page 4, 20-23, 25, #49, #74 seemed not appropriate.

#67 is too old; is the ref by the Authors the #42? At page 6 is missed at the end of paragraph 7.

·           It is not necessary to highlight how ‘important ‘ would be to study RUNX-1 in ovarian cancer. Instead, this concept could be better developed in the last paragraph.

·           He paragraph about ‘clinical therapy’ reports some not completely correct information about up-to-date therapeutic approach for ovarian cancer. Furthermore, the use of Bevacizumab in these tumors should be correctly reported.

·           Some English editing is required

See above

Author Response

1. Reference errors were checked and revised.

2: For the problems mentioned by the reviewer regarding the duplication of the previous and subsequent contents, the relevant contents have been deleted and modified.

3: Reference 42 is not the author's article, reference 88 is the article published by the author. At the same time, the author's research on ovarian cancer has been submitted, and the author has certain research support in the relevant conclusions.

4: Regarding the reviewer's proposal to review about RUNX1 target genes and biological function in leukemia, then in solid tumors thus going to ovarian cancer, the author makes a reply: First of all, the reviewer's suggestion has a progressive logic, and the author confirms it. However, considering the special issue of Signaling Pathways in Gynecological Malignant Tumors, the authors will focus on the roles of RUNX1 in the regulation of signaling pathways in ovarian cancer in order to better fit the characteristics of the special issue. At the same time, the role of leukemia and solid tumors is also described in this paper. Therefore, the authors did not review leukemia and solid tumors too much.

5: This article introduces the bevacizumab strain in order to prove the existence of combination drugs can achieve better therapeutic effect in clinical practice, thus indicating that RUNX1 has the potential to be used as a combination therapy target.

Round 2

Reviewer 1 Report

Please make sure that all the references are cited in the article.

Author Response

According to the comments of the reviewers, the authors checked whether all references were cited in the article and confirmed that they were cited in the article.